# Hemoadsorption with CytoSorb in Septic Shock Reduces Catecholamine Requirements and In-Hospital Mortality: A Single-Center Retrospective ‘Genetic’ Matched Analysis

**DOI:** 10.3390/biomedicines8120539

**Published:** 2020-11-26

**Authors:** Christopher Rugg, Riko Klose, Rouven Hornung, Nicole Innerhofer, Mirjam Bachler, Stefan Schmid, Dietmar Fries, Mathias Ströhle

**Affiliations:** Department of Anesthesiology and Critical Care Medicine, Medical University of Innsbruck, Anichstrasse 35, 6020 Innsbruck, Austria; christopher.rugg@tirol-kliniken.at (C.R.); riko.klose@student.i-med.ac.at (R.K.); rouven.hornung@tirol-kliniken.at (R.H.); nicole.innerhofer@tirol-kliniken.at (N.I.); mirjam.bachler@tirol-kliniken.at (M.B.); stefan.schmid@tirol-kliniken.at (S.S.); dietmar.fries@i-med.ac.at (D.F.)

**Keywords:** sepsis, septic shock, blood purification, hemoadsorption, CytoSorb

## Abstract

Septic shock is a major burden to healthcare with mortality rates remaining high. Blood purification techniques aim to reduce cytokine levels and resultant organ failure. Regarding septic shock, hemoadsorption via CytoSorb seems promising, but the main effects on organ failure and mortality remain unclear. In this retrospective single-center study, septic shock patients receiving CytoSorb in addition to renal replacement therapy (n = 42) were analyzed and compared to matched controls (n = 42). A generalized propensity-score and Mahalanobis distance matching method (‘genetic’ matching) was applied. Baseline comparability was high. Differences were merely present in higher initial Sequential Organ Failure Assessment (SOFA) scores (median and interquartile range: 13.0 (12.0–14.75) vs. 12.0 (9.0–14.0)) and requirements of norepinephrine equivalents (0.54 (0.25–0.81) vs. 0.25 (0.05–0.54) µg/kg/min) in the CytoSorb group. While remaining fairly constant in the controls, the catecholamines decreased to 0.26 (0.11–0.40) µg/kg/min within 24 h after initiation of CytoSorb therapy. In-hospital mortality was significantly lower in the CytoSorb group (35.7% vs. 61.9%; *p* = 0.015). Risk factors for mortality within the CytoSorb group were high lactate levels and low thrombocyte counts prior to initiation. Hereby, a cut-off value of 7.5 mmol/L lactate predicted mortality with high specificity (88.9%). Thus, high lactate levels may indicate absent benefits when confronted with septic shock patients considered eligible for CytoSorb therapy.

## 1. Introduction

In 2016, sepsis and septic shock were redefined [1]. Focus shifted from deranged vital signs (fever, tachycardia, tachypnoea) accompanied by leucocytosis or -penia to the occurrence of a life-threatening organ dysfunction due to a pathological host response to an infection. Despite the increase in diagnostic accuracy and the implementation of international guidelines and care bundles [2], sepsis and septic shock related mortality remains high and fairly unchanged since 2011 [3,4]. Pathophysiological motor powering the inappropriate host response is thought to be excessive cytokine release due to the overwhelming inflammation [5]. Cytokine genes are upregulated [6] and their release can induce increased production and further release, feeding a vicious cycle [7]. Importantly, pro- and anti-inflammatory mechanisms are triggered simultaneously, potentially also causing collateral damage in both directions (e.g., cell/tissue/organ damage, immune paralysis) [5,8,9]. Underlining clinical importance, combined high levels of pro- and anti-inflammatory cytokine activity rather than a disbalance between the two has been linked to a clearly increased mortality in septic patients [10,11,12,13]. This may partially explain why approaches targeting singular proinflammatory pathways might fail, and also why nonspecific, broad removal of cytokines via extracorporeal blood purification seems tempting [14].

Aiming to restore immune-homeostasis, different approaches to extracorporeal blood purification have evolved including conventional renal replacement therapy, high volume hemofiltration, high cut-off membranes, and adsorption techniques [15], where the hoped for results are an amelioration of organ failure and mortality. Regarding sepsis, up to now, only plasma exchange and hemoadsorption appear to be potentially effective [16], but due to very low confidence in the evidence, no recommendations have been made in the latest international guidelines [2]. Two main models have been hypothesized regarding the mode of action of blood purification techniques. On one hand, the cytotoxic theory is based upon the attenuation of peak concentrations [5,14,17,18]; and on the other, the cytokinetic theory relies on the restoration of cytokine gradients between blood and affected tissues by cytokine removal [19,20].

Given the fact that most inflammatory mediators are of middle-molecular-weight (5–60 kDa) [21], the use of blood purification techniques capable of removing such molecules seems obvious. By targeting within this range of molecular-weight, the CytoSorb hemoadsorption columns (CytoSorbents Corporation, Monmouth Junction, NJ, USA) are able to remove a broad spectrum of pro- and anti-inflammatory cytokines as well as damage- and pathogen-associated-molecular-patterns [22,23]. These special 300 mL cartridges consist of biocompatible, nonpolar, highly porous, polymer beads of 300–800 µm diameter [24]. The comparatively large total surface area of more than 40,000 m^2^ leads to potentially high substance removal by nonspecific surface absorption. Hereby, substances are removed in a concentration dependent manner. Highly concentrated substances are cleared more effectively while substances with low plasma levels are removed to a lesser extent, therefore adding to application safety [8,19,25,26]. Prospective data on CytoSorb use in sepsis are scarce and either not randomized or not powered for mortality [27,28,29]. Data from animal studies [26,30], case reports [31,32,33,34], and retrospective observational studies [35,36,37,38] predominantly present promising results. A recent retrospective, propensity-score matched study was able to show a decrease in all-cause 28-day mortality in the group treated with CytoSorb and renal replacement therapy (RRT) when compared to treatment with RRT alone [39]. Due to greater differences in group baseline characteristics, significant mortality differences were merely detected after special statistical weighting. The methodology has been challenged, therefore possibly limiting the value of the study [40].

We aimed to further elucidate the effects of CytoSorb use in septic shock with regard to organ failure, especially catecholamine requirements and mortality. Furthermore, risk factors for mortality within the group treated with CytoSorb were evaluated.

## 2. Materials and Methods

This retrospective study was approved by the Ethics Committee of the Medical University of Innsbruck (No.: 1124/2018, approval date: 15 October 2018) and the Institutional Review Board and registered with Clinical Trials (NCT04567199). Due to its retrospective design, consent to participate was not applicable.

### 2.1. Study Location

The study was conducted at the two general and surgical intensive care units (ICUs) of the department for anesthesiology and critical care medicine of the Innsbruck Medical University Hospital. In 23 level-three beds, annually 650–700 patients following scheduled and emergency cardiac, vascular, thoracic, abdominal, and traumatological surgery as well as transplantations and patients suffering from primary or secondary (postoperative) sepsis are treated.

### 2.2. Study Population

Patients treated with CytoSorb between 1 January 2015 and 31 December 2019 were obtained from our documentation database. CytoSorb treatment is mainly, but not exclusively indicated for septic shock in our department and is always combined with renal replacement therapy. By including patients only receiving CytoSorb treatment due to septic shock, n = 42 patients were enrolled. Inclusion criteria was septic shock requiring RRT on ICU, regardless of sepsis source and time of outbreak. Admission diagnoses were not restricted to any kind of sepsis, leading to the inclusion of patients suffering from primary as well as secondary sepsis. Initiation of CytoSorb therapy varied from 0.5 to 719 h after ICU admission, but the majority of patients received treatment within the first days (n = 20 on day 1, n = 7 on day 2, n = 6 on day 3, n = 1 on days 4, 5, 7, 9, 12, 27, and 31, and n = 2 on day 8) leading to a median time of CytoSorb mounting of 21.4 h after ICU admission. As we sought to not only compare baseline characteristics and outcome but also the course of certain variables during ICU stay (including catecholamine requirements and organ failure), baseline for the CytoSorb patients was defined as the day of CytoSorb mounting. In order to unitize the matched controls, we decided to define their baseline to be the time of ICU admission and searched for matched controls as deteriorated on ICU admission as the CytoSorb patients were just before CytoSorb initiation. Matched controls were treated at our department for septic shock and required RRT, but did not receive CytoSorb therapy. Again, admission diagnoses were not restricted to primary or secondary sepsis of any source. Anticipating a low number of patients as deteriorated directly on ICU admission as CytoSorb patients were directly before CytoSorb initiation, the searched treatment timeframe for the control group was extended back to 2011, where CytoSorb was not available yet. Despite this extension, the number of matching controls were limited, rendering an enlarged 1:n matching process impossible.

Data were obtained from a database containing all patients treated at our department within a timeframe from 1 November 2011 to 31 December 2019 (n = 2936). The primary data in this study contained basic demographics and laboratory values but did not include diagnoses nor catecholamine- or oxygen-requirements or sequential organ failure assessment (SOFA) scores. Therefore, matching took place for age, gender, baseline-bilirubin, -creatinine, -c-reactive-protein, -procalcitonin, -lactate, -thrombocyte-, and -leukocyte-count. In a second step, the resulting matches were analyzed for an admission diagnosis of septic shock and the requirement of renal replacement therapy in the local hospital information system. Matches missing the above-mentioned criteria were discharged and the matching process repeated for the remaining patients. The control group allocation process is depicted in Figure A1.

### 2.3. Data Acquisition

For the resulting study population (n = 82), an additional analysis of the ICU patient data management system and the hospital information system was conducted. Data obtained included the date and time of CytoSorb mounting, number of CytoSorb treatments received, requirement of mechanical ventilation and renal replacement therapy, fraction of inspired oxygen, arterial partial pressure of oxygen, mean arterial pressure, application and dosage of vasoactive and inotropic drugs (norepinephrine, epinephrine, vasopressin, milrinone, dobutamine), and ICU as well as hospital length of stay (LOS), date of death if applicable, main ICU diagnosis, sepsis location, (suspected) sepsis source, and existing comorbidities. Derived values included SOFA score, 28-day-mortality, in-hospital-mortality, and time from ICU admission to CytoSorb mounting. The SOFA score was calculated in accordance to the review by Lambden et al. in 2019 [41]. Therefore, the cardiovascular component was set utilizing norepinephrine equivalents that they proposed. In doing so, the use of vasopressin was accounted for more correctly (0.04 U/min vasopressin equates to 0.1 µg/kg/min norepinephrine equivalent). Due to incomplete documentation of Glasgow-Coma-Scales (GCS) prior to sedation and intubation as well as during ICU stay, a value of 15/15 was suspected throughout the study population, as also proposed by Lambden et al. [41]. The renal component was set to 4 points during phases of RRT and adjusted when RRT was not further required.

### 2.4. Statistics

Data were primarily stored in Excel (v16.4, Microsoft, Redmond, WA, USA). After importing, further statistical analysis was performed utilizing R (v4.0.2, R Core Team, www.R-project.org) and RStudio (v1.2.5001, RStudio, Inc., Boston, MA, USA).

Matching was performed by using the ‘Matching’ package for R [42]. This package gives the possibility of performing matching not only via the classical methods of propensity score or Mahalanobis distance matching, but also by the matching algorithm named ‘genetic’ matching (GenMatch), which is a generalization of the two, aiming at the maximization of covariate balances between the treated and control groups [42,43]. While the common approach of propensity score matching requires ellipsoidal distribution (e.g., normal) of the covariates (which is seldomly met) as well as knowledge or at least proper estimation of the propensity score model, the genetic matching algorithm of GenMatch is independent of the above-mentioned. As already shown, matching via the genetic algorithm GenMatch outperformed matching via propensity score, especially in terms of a lower mean square error [42].

The final matching results were analyzed by verifying balance within demographics and other baseline values by testing for treatment effect with regard to 28-day- and in-hospital-mortality via the McNemar test and by estimating the probability of treatment with CytoSorb within the complete cohort via multivariate logistic regression analysis (age, gender, pre-existing comorbidities, sepsis source, pathogen and baseline lactate levels, norepinephrine equivalents, and SOFA score).

When comparing the two groups consisting of multiple intraindividual longitudinal data over time, a modified analysis of variance (ANOVA) for nonparametric data was applied (’nparLD´package for R) [44]. Survival analysis was performed via Kaplan–Meier curves (‘Survminer’ and ‘Survival’ package for R) and corresponding log-rank-test.

With regard to risk factors for in-hospital mortality within the CytoSorb group, univariate logistic regression analysis, followed by a multivariate analysis, was performed. Due to a small sample size, decision was made to only include variables to the multivariate analysis with a *p* < 0.05 in the univariate analysis. As collinearity is present between SOFA score and thrombocyte count, only the latter was added to the model. Results are presented as odds ratio (OR) and 95% confidence interval (CI). Receiver operating characteristics (ROC) analysis via Youden-index maximization was conducted to determine the cut-off values for the baseline lactate levels and thrombocyte counts to predict mortality within the CytoSorb group.

In general, data are presented as count and percentage or due to non-normal distribution as median and interquartile range (IQR) where applicable. Regarding demographics, Fisher’s exact test was performed to detect group differences in frequencies and the Mann–Whitney U-test for group differences of continuous data. A *p*-value < 0.05 was considered significant.

## 3. Results

### 3.1. Baseline Characteristics

General demographics and baseline characteristics are itemized in Table 1. The matching process resulted not only in good balance between groups regarding matched covariates (age, gender, bilirubin, creatinine, thrombocytes, leukocytes, c-reactive protein (CRP), procalcitonin (PCT), lactate), but also regarding suspected pathogen, sepsis source, and comorbidities. Particularly with regard to the total number of pre-existing comorbidities per patient, non-survivors suffered from more comorbidities than survivors did in both groups, but there were no detectable differences between the groups. Additionally, ICU- and in-hospital LOS were comparable. Treatment effect of CytoSorb was verified by the McNemar test with regard to 28-day- (*p* = 0.029) and in-hospital mortality (*p* = 0.015). Together with a significantly higher requirement in norepinephrine equivalents and a nearly significant lower fraction of inspired oxygen (FiO_2_) to arterial oxygen partial pressure (PaO_2_) ratio, median SOFA scores were higher at baseline in the CytoSorb group (13 vs. 12). Uni- and multivariate logistic regression analysis, conducted to determine the probability of receiving CytoSorb therapy within the complete cohort, revealed that neither age nor gender, pre-existing comorbidities, pathogen, sepsis source, nor baseline lactate levels and norepinephrine requirements were significant contributors. Merely an increased SOFA score at baseline increased the odds of receiving CytoSorb within our matched cohort (OR: 1.26; 95% CI: 1.01–1.56; *p* = 0.042).

CytoSorb was mounted in median 21.4 h after ICU admission (IQR: 8.4–61.9 h; range: 0.5–719.0 h). While 38 of 42 CytoSorb patients received one single treatment, n = 3 received two and one single patient received a total of six CytoSorb treatments. Standard duration of one CytoSorb treatment is 24 h in our department.

### 3.2. Development of Catecholamine Requirements and Organ Failure

Catecholamine requirements are depicted as norepinephrine equivalents as proposed by Lambden in 2019 (Figure 1) [41]. Due to the study design, catecholamine requirements before baseline (ICU admission vs. CytoSorb initiation) were not available for the control group. Following a steep incline prior to the mounting of CytoSorb, a steep decline could be seen afterward in the CytoSorb group (Figure 1). While peak median values reached 0.55 µg/kg/min just prior to CytoSorb mounting (hour −1), median norepinephrine equivalents halved to 0.26 µg/kg/min within 24 h thereafter. Significantly reduced total catecholamine requirements can be seen in the Match group, with the greatest difference within the first hour referring to the baseline (Table 1, Figure 1). Unlike the CytoSorb group, no steep in- nor decline can be seen within the first 48 h.

The course of daily SOFA scores is illustrated in Figure A3 and shows no difference between the groups considering the entire inspected timeframe (day 1–28). Daily developments of other surrogates for organ failure between day 1 and 28 were analyzed in Figure A3. No significant differences were detected, despite a more pronounced recovery of thrombocyte count, but also higher bilirubin levels in the CytoSorb group.

### 3.3. Mortality

In-hospital- as well as 28-day-mortality differed significantly between the groups (35.7% vs. 61.9% and 21.4% vs. 47.6%, respectively; Table 1). Referring to initial (baseline) SOFA scores, mean predicted mortality was 85.1% in the CytoSorb and 67.5% in the Match group. Analysis of in-hospital survival via Kaplan–Meier curves revealed a nearly parallel course between groups before ICU day 7 and after ICU day 21 (Figure 2). However, between ICU day 7 and 21, a steeper decline in survival could be seen in the Match group.

### 3.4. Risk Factors for Mortality within the CytoSorb Group

Comparing survivors to non-survivors within the CytoSorb group, Figure 3 presents an analysis of surrogates of disease with a potential relation to severity. With catecholamine requirements, bilirubin, lactate, and procalcitonin levels remained higher over time in non-survivors, and survivors presented with a significantly higher recovery of thrombocyte count. Regarding baseline values just prior to CytoSorb mounting, thrombocytes were significantly higher in survivors (130 (101–243) G/l vs. 74 (59–119) G/l; *p* = 0.007) and initial lactate levels were clearly higher in non-survivors (7.6 (2.6–11.1) mmol/L vs. 2.8 (1.7–4.9) mmol/L; *p* = 0.036).

Univariate logistic regression analysis regarding in-hospital mortality within the CytoSorb group revealed higher age, presence of abdominal sepsis, pre-existing chronic kidney disease, elevated initial SOFA scores, and lactate levels as well as a decreased thrombocyte count as risk factors (Table 2). Fitting age, presence of abdominal sepsis, pre-existing chronic kidney disease, initial thrombocytes, and lactate into a multivariate logistic regression model left merely initial lactate levels and thrombocyte counts to remain with significance (OR 1.27 (1.00–1.60) per mmol/L and OR 0.98 (0.96–-1.00) per G/L). ROC-curve analyses with Youden index maximization determined a lactate level higher than 7.5 mmol/L (sensitivity: 53.3%; specificity: 88.9%) and a thrombocyte count less than 99 G/L (sensitivity: 66.7%; specificity: 81.5%) prior to CytoSorb mounting to predict mortality within the CytoSorb group.

## 4. Discussion

In this retrospective, single-center study, patients treated with CytoSorb and renal replacement therapy for septic shock were analyzed and compared to a matched cohort without CytoSorb therapy, but with an admission diagnosis of septic shock and requirements for renal replacement therapy. Aside from clearly elevated catecholamine requirements and a slightly, but significantly higher SOFA score in the CytoSorb group at baseline, general demographics, sepsis sources, suspected pathogens, pre-existent comorbidities, and other baseline characteristics were comparable between the groups. Median catecholamine requirements approximately halved within 24 h after CytoSorb mounting. In-hospital as well as 28-day-mortality were significantly lower in the CytoSorb group when compared to the Match group. Main differences in the survival curves took place between days 7 and 21. Within the CytoSorb group, elevated lactate levels and decreased thrombocyte counts were significant risk factors for mortality. Hereby, lactate levels above 7.5 mmol/L prior to CytoSorb mounting predicted mortality with high specificity.

### 4.1. Baseline Characteristics

Up to now, only a few controlled studies regarding CytoSorb therapy in septic shock have been published. In 2019, Brouwer et al. were able to show a significant decrease in observed compared to SOFA score predicted mortality [39]. Furthermore, after adjusting for significant differences in baseline characteristics, a decreased mortality was shown for the CytoSorb-treated group. Aside from these intergroup discrepancies, foremost including age and pre-existing comorbidities but also sepsis source, lactate levels, norepinephrine requirements, and SOFA scores at the start of therapy, the group sizes also differed. Due to this, the methodology of the study has been questioned [40]. In the presented study, the chosen matching process yielded one best possible match per patient resulting in two equally sized groups with sufficient comparability between age, gender, sepsis source, suspected pathogen, comorbidities, and most observed baseline values. Merely SOFA score and norepinephrine requirements at baseline differed toward higher values in the CytoSorb group. With missing guidelines toward adjunctive hemoadsorptive therapies in septic shock, their use remains indicated by the beliefs of the intensivist in charge, mostly as salvage therapy for the most severely affected only. Then again, patients with a high severity of illness also seem to benefit most [28,37,45], which has led to recommendations toward this subgroup of patients when indicating CytoSorb therapy [8]. The high initial SOFA scores, catecholamine requirements, lactate levels, and observed mortalities in this and also other studies on CytoSorb treatment in septic shock are in line with these statements [32,38,39]. In the median, time to CytoSorb mounting was within the first 24 h, which seems to be beneficial, as supported by increasing literature [28,37,45,46]. Treatment length of one cartridge was 24 h in this study and most patients received one cartridge only. Another study also changed cartridges every 24 h, but treated for a minimum of three days [37] and again, and one study treated for 6 h a day for seven days [27]. While the when seems clearer (within 24 h), the how long and how often still remains greatly debatable regarding CytoSorb [8].

### 4.2. Development of Catecholamine Requirements and Organ Failure

In theory, blood purification techniques aim to ameliorate cytokine load and resultant organ dysfunction. In practice, cytokines are removed, but the true effects on organ function remain unclear [47]. Aside from cytokines, bile acids, myoglobin and free hemoglobin are also removed, potentially being beneficial in liver and kidney injury as well as vascular tone and microvascular density [48,49]. Improved hemodynamics accompanied by a reduction in catecholamine requirements is probably the best described positive effect of CytoSorb up to date [28,35,37,45,50] and has been acknowledged as a relevant endpoint, possibly affecting other organ dysfunction [8]. By recording the needs of norepinephrine equivalents [41] before and after the initiation of CytoSorb therapy, an initial deterioration prior to mounting, followed by a clear stabilization within 24 h thereafter could also be seen within our data. Compared to the matched controls, an initially significant difference seen at the baseline (Table 1, Figure 1) was followed by a convergence toward similar values within the two groups (Figure 1). Surprisingly, no significant differences in the development of SOFA scores, specific organ functions, or procalcitonin resulted regarding a timeframe of 28 days (Figure A2 and Figure A3). Furthermore, although cumulative vasopressor loads are expected to contribute to mortality in intensive care [51], 28-day and in-hospital mortality were higher in the Match group, with initially lower and later on comparable catecholamine requirements. Given these results, the observed decreased mortality cannot be deduced from CytoSorb derived reduction in vasopressor needs with the presented data. Further randomized trials including patients suffering from fulminant hemodynamic deterioration in septic shock are needed to clarify this issue.

### 4.3. Mortality

Sepsis and septic shock represent very heterogeneous entities. Depending on diverse factors (e.g., involved pathogen, pathogen load, virulence, sepsis source, general immune response, and other intrinsic factors), the trajectories can differ greatly. General mortality in septic shock has been described in ranges from 15 to 56% [3,9,52,53]. A very recent meta-analysis stated rates of 34.7% for 30-day and 38.5% for 90-day mortality in septic shock in Europe, North America, and Australia [4]. More importantly, they did not observe any changes in mortality over time since 2011. This is of special interest as patients allocated to the Match group in this study were treated between 2011 and 2019, whereas patients from the CytoSorb group were treated between 2015 and 2019. Intergroup mortality differences merely due to time differences are therefore highly improbable. Regarding CytoSorb, a registry analysis revealed an observed mortality of 65% compared to a predicted mortality of 78% in all patients treated for septic shock [46]. A meta-analysis of blood purification techniques in septic patients demonstrated a clear mortality difference between the treatment (35.7%) and the conventional therapy groups (50.1%) [16]. After adjusting for baseline discrepancies, the mentioned study by Brouwer et al. reported mortality rates of 53 vs. 72% when comparing the treatment with CytoSorb and treatment with RRT alone in septic shock [39]. With regard to the data presented by us, the mortality rate in the CytoSorb group was 35.7%, which was exactly as high as reported by the meta-analysis [16] and far below the predicted mortality. Mortality within the Match group ranged between the above described studies (61.9%) and was merely somewhat lower than predicted by SOFA score. Hereby, mortality prediction was based on a SOFA score assuming a GCS of 15/15, therefore rather underestimating true mortality. Analyzing the Kaplan–Meier curves from ICU admission to hospital discharge revealed an interesting finding. Although CytoSorb therapy was mainly initiated within the first day and merely applied for 24 h, main survival differences occurred between days 7 and 21. Early fulminant deaths within the first days were equal in both groups, and so was the decline of survival after day 21. With missing differences in surrogates of organ function between the groups and also missing measurements of cytokine levels or other markers of immune-competency, we can only assume the origin of this observation. Better, faster, or merely more adequate restoration of immune-homeostasis is certainly an alluring theory in this case, as its absence in the Match group may have led to a state of persistent inflammation and immunosuppression as proposed in different settings [8,54]. Subsequently, this state may have led to more secondary complications (e.g., infections), absence of recovery, and finally, an increased mortality. Particularly regarding the timeframe between days 7 and 21, there are hints on increased norepinephrine requirements and lactate levels as well as decreased thrombocytes and PaO_2_ to FiO_2_ ratios in the Match group (Figure A3). Exact mechanisms of mortality reduction remain unclear.

### 4.4. Risk Factors for Mortality within the CytoSorb Group

Literature analyzing who will and particularly who will not profit from CytoSorb therapy is scarce, but urgently needed. A prospective study assessing interleukin (IL)-6 levels with or without CytoSorb therapy in acute respiratory distress syndrome (ARDS) patients found decreased IL-6 levels, but no survival benefit in the CytoSorb group [27]. The propensity-score matched analysis by Brouwer et al. also found that pulmonary sepsis was associated with higher odds of dying in the CytoSorb treated group [39]. Aside from older age, higher lactate levels at baseline also remained risk factors for mortality in their multivariate model. We also conducted a univariate followed by a multivariate analysis with regard to mortality within the CyotSorb group. While older age, existing abdominal sepsis, and pre-existing chronic kidney failure lost significance in the multivariate model, higher baseline lactate levels and lower baseline thrombocyte counts remained significant. Of importance is that CytoSorb treatment initiation itself is biased, as treatment is mainly withheld for cases with fulminant progression but presumed improvability. We assumed this indication bias to be the reason why neither age nor gender nor comorbidities turned out to be significant risk factors for fatal outcome in the analysis of our CytoSorb group. We also believe though that the clear signal we detected from the baseline lactate levels is therefore of even more importance, especially as it has been previously confirmed in the study by Brouwer et al. [39]. Due to the large difference between the baseline lactate levels in survivors and non-survivors, we were able to determine a cut-off value to predict mortality prior to CytoSorb initiation with high specificity.

### 4.5. Limitations

As already mentioned, limitations include possible confounding by indication as well as missing data on cytokine levels. Furthermore, data on RRT-setup, fluid management, ventilatory, and antibiotic management were not included in the analysis. However, due to the single-center setup in this study, similar strategies in case management can be assumed. Despite great effort put into the matching process, there are some concerns due to our approach. First, selection bias cannot be completely excluded as final proof of matching took place manually primarily due to missing data on admission diagnoses and RRT requirements. Second, general definition of baseline differed between the groups. Initiation of CytoSorb therapy was not persistent within the first day of admission, wherefore decision was made to match patients regarding day of CytoSorb mounting in the CytoSorb and day of ICU admission in the Match group. Limitations in comparability due to possible differences in existing disease stages is also a bias in mortality, where matched controls die before even having the possibility of CytoSorb treatment could be feared. However, as seen in the Kaplan–Meier curves, mortality in the Match group mainly took place after day 7, diminishing concerns on early mortality bias. Furthermore, due to the matching process, the controls presented as deteriorated on ICU admission as did patients prior to CytoSorb initiation with regard to the matched variables (age, gender, bilirubin, creatinine, CRP, PCT, lactate, platelet, and leukocyte count). The presented similar course of SOFA scores and specific markers of inflammation and organ dysfunction between the groups also rule out greater differences regarding disease stage. Third, our analysis included patients suffering from community and hospital acquired as well as primary and secondary sepsis. Particularly within the CytoSorb group, patients were included with treatment initiation far beyond ICU admission. Comparability is therefore potentially hindered. In this regard, we must state that thee inclusion criteria were not restricted to any kind of sepsis in either group. ICU admission alone does not necessarily indicate any particular sepsis type. Most importantly, sepsis source and suspected pathogens were comparable between the groups with nearly half being Gram-negative infections in both groups. Greater intergroup differences with regard to hospital or community acquired infections can again be ruled out as different pathogens would be expected. Fourth, the included treatment period differed between the groups (2015 vs. 2011–2019), further raising concerns due to possible improvement in mortality over time. This practice was necessary in order to find an adequate number of matched controls as deteriorated on ICU admission as patients prior to CytoSorb initiation and also explains why a greater sample size with 1:n matching was not possible. Noteworthy in this regard, the recent meta-analysis by Bauer et al. in 2020 showed that mortality in septic shock has not improved since 2011 [4].

Finally, although both groups were comparable with regard to demographics, most baseline characteristics and disease etiology, differences were still present with higher initial SOFA scores and catecholamine requirements in the CytoSorb group. Nevertheless, our results are in accordance with existing results from other observational and animal studies as well as many case reports. In general, the retrospective design within one single-center limits the validity of the study, and once more calls for well-designed randomized trials in order to confirm the presented results.

## 5. Conclusions

The addition of extracorporeal hemoadsorption via CytoSorb to standard care in septic shock patients requiring renal replacement therapy approximately halved catecholamine requirements within 24 h. In-hospital as well as 28-day mortality were reduced significantly when compared to a generalized propensity-score and Mahalanobis distance matched group. When confronted with septic shock patients considered eligible for CytoSorb therapy, high lactate levels may indicate absent benefits.

## Figures and Tables

**Figure 1 biomedicines-08-00539-f001:**
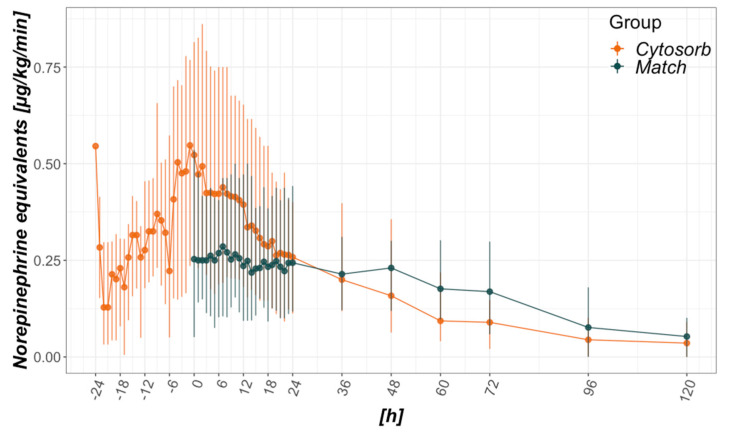
Catecholamine requirements in relation to CytoSorb mounting (CytoSorb group) or intensive care unit (ICU) admission (Match group). Data presented as median and interquartile range. Norepinephrine equivalents are calculated as proposed in [41]. Regarding vasopressin 0.04 U/min equated to 0.1 µg/kg/min norepinephrine equivalents. The *x*-axis refers to hours from baseline, where 0 is time of CytoSorb mounting in the CytoSorb and time of ICU admission in the Match group.

**Figure 2 biomedicines-08-00539-f002:**
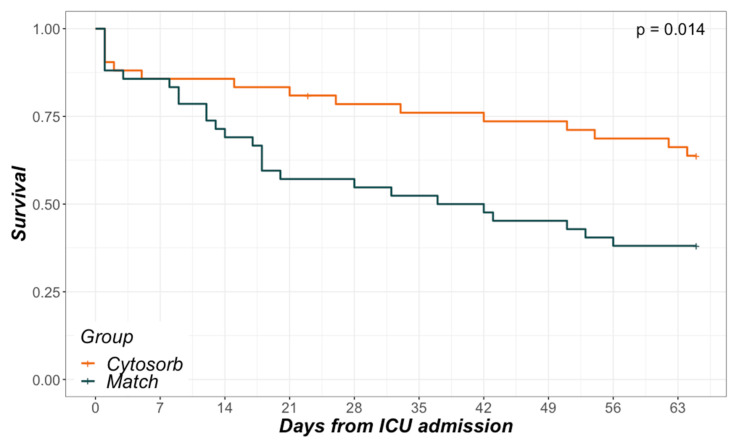
Kaplan–Meier curves. In-hospital survival in relation to days from ICU admission.

**Figure 3 biomedicines-08-00539-f003:**
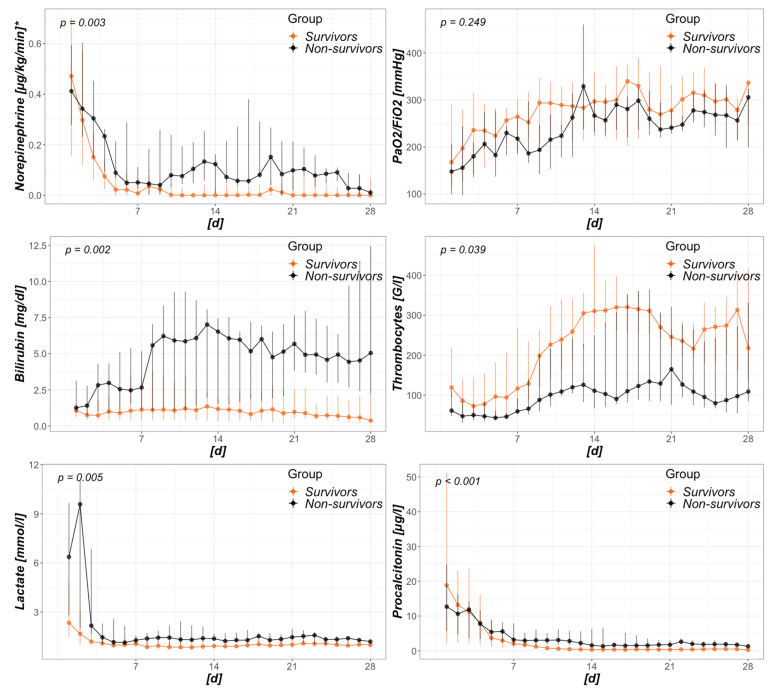
Intensive care unit (ICU)-day dependent analysis of survivors and non-survivors treated with CytoSorb. Data presented as daily median and interquartile range. The *x*-axis refers to days from baseline, where first day is that of CytoSorb mounting. * displayed as norepinephrine equivalents (in accordance to [41]).

**Table 1 biomedicines-08-00539-t001:** Demographics, baseline characteristics, and outcome.

	CytoSorb (n = 42)	Match (n = 42)	*p*
	n (%) or Median (IQR)	n (%) or Median (IQR)	
**Sex**			
**Male**	27 (64.3)	25 (59.5)	
**Female**	15 (35.7)	17 (40.5)	0.823
**Age**	64 (48–73)	68 (55–74)	0.400
**Sepsis source**			
**Abdominal**	15 (35.7)	15 (35.7)	1.000
**Pulmonal**	15 (35.7)	14 (33.3)	0.820
**Urogenital**	2 (4.8)	4 (9.5)	0.6760
**Soft tissue**	8 (19.0)	8 (19.0)	1.000
**Other**	9 (21.4)	6 (14.3)	0.569
**Suspected pathogen**			0.887
**Gram negative bacteria**	19 (45.2)	18 (42.9)	
**Gram positive bacteria**	17 (40.5)	15 (35.7)	
**Influenza**	2 (4.8)	1 (2.4)	
**Candida spp.**	1 (2.4)	2 (4.8)	
**Aspergillus spp.**	0 (0.0)	1 (2.4)	
**Unknown**	3 (7.1)	6 (14.3)	
**Comorbidities**	35 (83.3)	37 (88.1)	0.757
**Arterial hypertension**	32 (76.2)	30 (71.4)	0.804
**Cerebrovascular disease**	8 (19.0)	10 (23.8)	0.791
**COPD ^+^**	4 (9.5)	8 (19.0)	0.350
**Coronary artery disease**	15 (35.7)	16 (38.1)	1.000
**Diabetes mellitus type 2**	19 (45.2)	14 (33.3)	0.372
**Heart failure**	7 (16.7)	7 (16.7)	1.000
**Chronic kidney disease**	16 (38.1)	14 (33.3)	0.820
**Peripheral artery disease**	2 (4.8)	7 (16.7)	0.1560
**Total number of comorbidities**			
**All**	2.5 (1.0–4.0)	2.0 (0.3–4.0)	0.9280
**Survivors**	1.0 (0.0–3.0)	1.0 (0.0–3.3)	0.9790
**Non-survivors**	4.0 (2.0–5.0)	3.0 (2.0–4.8)	0.5270
**Baseline values**			
**SOFA score**	13.0 (12.0–14.75)	12.0 (9.0–14.0)	0.023
**PaO_2_/FiO_2_ [mmHg]**	160 (104–287)	234 (126–294)	0.095
**Norepinephrine [µg/kg/min] ***	0.52 (0.25–0.81)	0.25 (0.05–0.54)	0.014
**Bilirubin [mg/dL]**	1.18 (0.72–2.02)	1.03 (0.78–1.76)	0.747
**Thrombocytes [G/L]**	111 (70–172)	106 (75–187)	0.964
**Creatinine [mg/dL]**	1.84 (1.17–2.74)	1.80 (1.29–2.69)	0.758
**Leukocytes [G/L]**	13.95 (6.60–20.88)	9.93 (6.11–15.26)	0.304
**CRP [g/dl]**	13.64 (7.95–19.57)	14.47 (10.40–24.25)	0.508
**PCT [µg/L]**	15.53 (2.35–33.24)	8.56 (2.05–38.60)	0.567
**Lactate [mmol/L]**	3.5 (1.8–7.3)	3.4 (2.0–5.3)	0.687
**ICU LOS [d]**	21 (12–33)	15 (8–26)	0.121
**Hospital LOS [d]**	30 (17–49)	30 (13–48)	0.505
**28d—mortality**			
**Dead**	9 (21.4)	20 (47.6)	0.029
**alive**	33 (78.6)	22 (52.4)	
**In hospital mortality**			
**Dead**	15 (35.7)	26 (61.9)	
**alive**	27 (64.3)	16 (38.1)	0.015

* displayed as norepinephrine equivalents (in accordance to [41]) within first hour; ^+^ Chronic obstructive pulmonary disease.

**Table 2 biomedicines-08-00539-t002:** Uni- and multi-variate logistic regression analysis. Risk factors for mortality within the CytoSorb group.

	Crude Odds Ratio (95% CI)	*p*	Adjusted Odds Ratio (95% CI)	*p*
**Age (per year)**	1.08 (1.02–1.15)	0.009	1.09 (0.97–1.22)	0.164
**Female Gender**	0.85 (0.23–3.21)	0.81		
**Sepsis source**			7.86 (0.94–65.59)	0.057
**Abdominal**	11.00 (2.42–49.91)	0.002		
**Pulmonal**	0.34 (0.08–1.51)	0.156		
**Urogenital**	2.00 (0.12–34.6)	0.634		
**Soft tissue**	1.20 (0.24–5.97)	0.824		
**Suspected pathogen**				
**Gram negative bacteria**	reference			
**Gram positive bacteria**	0.53 (0.12–2.27)	0.390		
**Comorbidities (any)**	4.00 (0.43–36.92)	0.221		
**Arterial hypertension**	7.00 (0.79–61.97)	0.080		
**Cerebrovascular disease**	1.10 (0.22–5.42)	0.907		
**COPD**	6.50 (0.61–69.14)	0.121		
**Coronary artery disease**	3.27 (0.86–12.35)	0.081		
**Diabetes mellitus type 2**	1.09 (0.31–3.88)	0.890		
**Heart failure**	1.44 (0.28–7.5)	0.667		
**Chronic kidney disease**	4.29 (1.12–16.44)	0.034		
**Peripheral artery disease**	>99 (0–∞)	0.992	1.46 (0.16–13.52)	0.741
**Baseline characteristics**				
**SOFA Score (per point)**	1.57 (1.07–2.3)	0.02		
**PaO2/FiO2 (per mmHg)**	1.00 (0.99–1.00)	0.757		
**Norepinephrine (per µg/kg/min) ***	0.70 (0.09–5.59)	0.737		
**Bilirubin (per mg/dL)**	1.15 (0.87–1.51)	0.334		
**Thrombocytes (per G/L)**	0.98 (0.97–1.00)	0.015	0.98 (0.96–1.00)	0.049
**Procalcitonin (per µg/L)**	1.00 (0.99–1.00)	0.287		
**Lactate (per mmol/L)**	1.21 (1.03–1.43)	0.019	1.27 (1.00–1.60)	0.045

* displayed as norepinephrine equivalents (in accordance to [41]).

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
