# Peer review of "Hemoadsorption with CytoSorb in Septic Shock Reduces Catecholamine Requirements and In-Hospital Mortality: A Single-Center Retrospective ‘Genetic’ Matched Analysis"

_biomedicines, 2020, doi:10.3390/biomedicines8120539_

Round 1
Reviewer 1 Report
Authors investigated the effects of CytoSorb in septic shock using matched analysis. It is important issue in sepsis, and interesting.
The followings should be discussed.
Major issues
- They investigated risk factors for mortality within the CytoSorb group and found lactate level is important factor. With this result, they stated that lactate levels help predict a possible benefit of CytoSorb therapy. Lactate level was independent factor for mortality in the CytoSorb group, so it is not reasonable to state that lactate levels could help the candidate for CytoSorb therapy. On the contrary, they could find risk factors for mortality in non-CytoSorb group, and state that if the patients have high risk, CytoSorb could be applied. Unfortunately, the latter also does not seem scientifical, so I recommend not to use the sentence that “lactate level help predict a possible benefit.”
- As they commented in limitation, the different time period between two groups are critical limitations even though they performed matched analysis. We could not match groups with un-revealed variables. However, this can’t be overcome with this design. They stated that mortality of sepsis in 2011 and 2020 are not different, but most clinicians could not feel like that. So I recommend this limitation should be remained main limitation (not to mention mortality rate could be same in whole study period).
Minor issues
- A little typos and grammatical errors are seen.
Author Response
Authors investigated the effects of CytoSorb in septic shock using matched analysis. It is important issue in sepsis, and interesting.
The followings should be discussed.
Major issues
- They investigated risk factors for mortality within the CytoSorb group and found lactate level is important factor. With this result, they stated that lactate levels help predict a possible benefit of CytoSorb therapy. Lactate level was independent factor for mortality in the CytoSorb group, so it is not reasonable to state that lactate levels could help the candidate for CytoSorb therapy. On the contrary, they could find risk factors for mortality in non-CytoSorb group, and state that if the patients have high risk, CytoSorb could be applied. Unfortunately, the latter also does not seem scientifical, so I recommend not to use the sentence that “lactate level help predict a possible benefit.”
Dear Reviewer,
we would like to thank you for your time and work with our manuscript. We are sorry for the misleading wording with regard to lactate levels in CytoSorb patients. As mentioned by you lactate levels were shown to be an independent risk factor for mortality within the CytoSorb group. By area under the ROC curve analysis baseline lactate levels (prior to CytoSorb initiation) were analysed for their quality of predicting mortality within the CytoSorb group. With high specifity (88.9%) lactate levels above 7.5mmol/l were shown to predict mortality despite CytoSorb treatment.
Analysis of risk factors for mortality within the non-CytoSorb group would have been possible but deduction of recommendations towards a possible benefit of CytoSorb treatment not possible with the chosen study design and acquired results. Again, only randomized controlled trials would be able to exactly determine which subgroup of patients could benefit when from CytoSorb treatment.
For better clarity we rephrased the mentioned sentence to the following:
Abstract page 1 line 28: “Hereby, a cut-off value of 7.5 mmol/l lactate predicted mortality with high specificity (88.9%). Thus, high lactate levels may indicate absent benefits when confronted with septic shock patients felt eligible for CytoSorb therapy.”
and Conclusion page 11 line 32: “When confronted with septic shock patients felt eligible for CytoSorb therapy, high lactate levels may indicate absent benefits.”
- As they commented in limitation, the different time period between two groups are critical limitations even though they performed matched analysis. We could not match groups with un-revealed variables. However, this can’t be overcome with this design. They stated that mortality of sepsis in 2011 and 2020 are not different, but most clinicians could not feel like that. So I recommend this limitation should be remained main limitation (not to mention mortality rate could be same in whole study period).
Dear Reviewer,
Thank you for your valuable thoughts!
We do agree that different time periods of treatment between the groups is potentially able to cause substantial bias. This step was needed to perform accurate matching with regard to the mentioned variables. After matching, as seen in the table below, only few patients actually originated as far back as 2011 or 2012. Most of them were treated after 2014. Mortality in the Match group is also presented per year in the table below. Due to very small numbers, fractions must be seen with caution of course, but by these means no clear trend can be seen.
This would be in line with the recently published meta-analysis and cited reference:
Bauer, M.; Gerlach, H.; Vogelmann, T.; Preissing, F.; Stiefel, J.; Adam, D. Mortality in sepsis and septic shock in Europe, North America and Australia between 2009 and 2019— results from a systematic review and meta-analysis. Crit. Care 2020, 24, 239, doi:10.1186/s13054-020-02950-2
They found sepsis mortality to decrease over time but septic shock mortality to remain high!
They stated:
“Average 30-day septic shock mortality was 34.7% (95% CI 32.6–36.9%), and 90-day septic shock mortality was 38.5% (95% CI 35.4–41.5%)”
And more importantly:
“A statistically significant decrease of 30-day septic shock mortality rate was found between 2009 and 2011, but not after 2011.”
In the end they concluded:
“Trends of lower sepsis and continuous septic shock mortality rates over time and regional disparities indicate a remaining unmet need for improving sepsis management.”
In order to prevent interference with clinical perceptions we decided to rephrase the sentence as follows:
Discussion page 11 line 12:
“Noteworthy in this regard, the recent meta-analysis by Bauer et al. in 2020 showed that mortality in septic shock has not improved since 2011 [4].”
|
year |
2011 |
2012 |
2013 |
2014 |
2015 |
2016 |
2017 |
2018 |
2019 |
|
No. of Patients in the Match Group |
1 |
3 |
4 |
6 |
6 |
4 |
7 |
3 |
8 |
|
Mortality in the Match group n (%) |
1 (100%) |
2 (66.7%) |
1 (25.0%) |
5 |
4 |
2 (50.0%) |
4 (57.1%) |
1 (33.3%) |
6 (75.0%) |
Minor issues
- A little typos and grammatical errors are seen.
Dear Reviewer,
We proofread the manuscript and corrected to the best of our knowledge!
Thank you very much for your support in improving our manuscript.
Reviewer 2 Report
The evidence presented in this manuscript makes a convincing case about the potential utility of this approach to the removal of medium molecular weight compounds including cytokines involved in inflammatory responses. Oddly, cytokines are not mentioned in the abstract, although they are discussed among bioactive components involved in responses to this treatment for septic shock events. The decrease in catecholamine requirements and utility of lactate as a predictor of mortality presented are clearly enough.
Biomedicines requires a statement on ethics for research involving human subjects, and that appears to have been overlooked. That is especially troubling in studies involving experimental treatments and frequently involving mortality.
The scientific English used in this article seems casual at times and is in need of careful proofreading. Examples:
- P 1 line 17 and again P 9 line 19 The “true effects” of hemoadsorption are discussed. Presumably this vague term refers to a physiological mechanism of action, but clarifying true effects implies the existence of fictitious or artifactual effects that complicate the interpretations of treatments. Clarification is needed.
- Page 1 line 22 and elsewhere Values such as SOFA scores and other parameters are presented parenthetically followed by other numbers in additional sets of parentheses. These presumably mean means and a range of recorded values but, that is not stated or explained anywhere as far as I can tell.
- Some improvement in language accuracy, word choice, and proofreading are recommended:
- Page 1 line 40 The colloquialism “viscous cycle” is used, probably with the intent of saying vicious cycle. By context, I doubt that the authors mean to be referring to viscosity.
- At several other points throughout the manuscript, casual language is used in cases in which more precise scientific English could be applied. As one examples, see page 9 lines 38-40 “what… who… when.. where… and how hard…”.
- Page 2 line 10 Should be a comma after evidence, to read “…in the evidence, no recommendations…”.
- Page 2 line 25, subject-verb disagreement. Says “data… is scarce”, should say data are scarce. References elsewhere in the manuscript appear to treat the word data as a plural.
- Page 5 line 8 reads “A in total significantly reduced catecholamine requirement…”, which is awkward and grammatically incorrect. Could be reworded to read “An in-total significantly reduced…” or perhaps clarified to read something like “Significantly reduced total catecholamine requirements…”.
- Page 8 line 9, the casual descriptor “requirements decreased distinctly…” is a true enough statement, but changes in requirements could be presented in terms of percentage change and degrees of statistical significance. The same casual word use appears in the Conclusions, page 11 line 24.
- Page 8 line 17 states “…where able to show”, must mean “…were able to show…”.
- Page 10 line 22 should read “…who will not profit…”.
- Page 11, lines 17 and 18 “…are in accordance with existing results…” would be a preferable word choice than “…to existing results”.
Author Response
The evidence presented in this manuscript makes a convincing case about the potential utility of this approach to the removal of medium molecular weight compounds including cytokines involved in inflammatory responses. Oddly, cytokines are not mentioned in the abstract, although they are discussed among bioactive components involved in responses to this treatment for septic shock events. The decrease in catecholamine requirements and utility of lactate as a predictor of mortality presented are clearly enough.
Dear Reviewer,
thank you very much for your precise reading and your helpful thoughts.
Unfortunately, plasma levels of cytokines were not measured and therefore not accessible to us. Due to these missing values, we decided against putting too much emphasis on this topic. We do believe in the paramount importance of cytokine and other bioactive component removal with regard to CytoSorb treatment though. Therefore, we decided to discuss this topic more in depth within the introduction.
Biomedicines requires a statement on ethics for research involving human subjects, and that appears to have been overlooked. That is especially troubling in studies involving experimental treatments and frequently involving mortality.
Dear Reviewer,
once more thank you for your support. Unfortunately, the statement got “lost in translation” when fitting the manuscript to the biomedicines template. We dearly apologize for this mistake!
We added the following statement to the top of the Methods section:
“This retrospective study was approved by the Ethics Committee of the Medical University of Innsbruck (Nr.:1124/2018) and the Institutional Review Board and registered with Clinical Trials (NCT04567199). Due to its retrospective design a consent to participate was not applicable.”
The scientific English used in this article seems casual at times and is in need of careful proofreading. Examples:
- P 1 line 17 and again P 9 line 19 The “true effects” of hemoadsorption are discussed. Presumably this vague term refers to a physiological mechanism of action, but clarifying true effects implies the existence of fictitious or artifactual effects that complicate the interpretations of treatments. Clarification is needed.
We understand your objection and adapted the sentence as follows:
Page 1 line 17: “Regarding septic shock hemoadsorption via CytoSorb seems promising, but main effects on organ failure and mortality remain unclear.”
- Page 1 line 22 and elsewhere Values such as SOFA scores and other parameters are presented parenthetically followed by other numbers in additional sets of parentheses. These presumably mean means and a range of recorded values but, that is not stated or explained anywhere as far as I can tell.
We are sorry for the misunderstanding!
The answer to your question is stated on page 4 line 17: “In general, data are presented as count and percentage or due to non-normal distribution as median and interquartile range (IQR) where applicable.”
For better clarity we added the following to the abstract page 1 line 22:
“Differences were merely present in higher initial SOFA-scores (median and interquartile range: 13.0 (12.0-14.75) vs. 12.0 (9.0-14.0)) and requirements of norepinephrine equivalents (0.54 (0.25-0.81) vs. 0.25 (0.05-0.54) µg/kg/min) in the CytoSorb group.”
- Some improvement in language accuracy, word choice, and proofreading are recommended:
- Page 1 line 40 The colloquialism “viscous cycle” is used, probably with the intent of saying vicious cycle. By context, I doubt that the authors mean to be referring to viscosity.
Fully agree! Thank you for your helpful reading!
Changed to “vicious cycle”.
- At several other points throughout the manuscript, casual language is used in cases in which more precise scientific English could be applied. As one examples, see page 9 lines 38-40 “what… who… when.. where… and how hard…”.
We must admit that we somewhat deliberately phrased this sentence a bit strikingly. We do understand your objection with regard to scientific language though.
We rephrased the sentence as follows: page 9 line 40: “Depending on diverse factors (e.g. involved pathogen, pathogen load, virulence, sepsis source, general immune response and other intrinsic factors) trajectories can differ greatly.”
- Page 2 line 10 Should be a comma after evidence, to read “…in the evidence, no recommendations…”.
Done! Thank you!
- Page 2 line 25, subject-verb disagreement. Says “data… is scarce”, should say data are scarce. References elsewhere in the manuscript appear to treat the word data as a plural.
Very accurate reading. Thank you very much! Done!
- Page 5 line 8 reads “A in total significantly reduced catecholamine requirement…”, which is awkward and grammatically incorrect. Could be reworded to read “An in-total significantly reduced…” or perhaps clarified to read something like “Significantly reduced total catecholamine requirements…”.
Once more we must thank you for your support! We gladly accepted your suggestion and changed the sentence as follows: page 5 line 8: “Significantly reduced total catecholamine requirements can be seen in the Match group, with the greatest difference within the first hour referring to baseline (Table 1, Figure 1).”
- Page 8 line 9, the casual descriptor “requirements decreased distinctly…” is a true enough statement, but changes in requirements could be presented in terms of percentage change and degrees of statistical significance. The same casual word use appears in the Conclusions, page 11 line 24.
Dear Reviewer,
We do understand your objection. A more precise description of the changes in requirements can be read in the results section (page 5 line 7).
With exact presentations in Table 1 and Figures 1, 3 and A3 (in the appendix) and also within the text in the results section, we tried to desist from presenting to many exact numbers in the discussion and conclusion section - for better readability.
For a more scientific description, we did change the sentences upon your request to the following:
Page 8 line 11: “Median catecholamine requirements approximately halved within 24 hours after CytoSorb mounting.”
And page 11 line 24: “The addition of extracorporeal hemoadsorption via CytoSorb to standard care in septic shock patients requiring renal replacement therapy approximately halved catecholamine requirements within 24 hours.”
Page 8 line 17 states “…where able to show”, must mean “…were able to show…”.
Done. Thank you very much!
- Page 10 line 22 should read “…who will not profit…”.
We apologize for this unfortunate mistake. Changes were done!
- Page 11, lines 17 and 18 “…are in accordance with existing results…” would be a preferable word choice than “…to existing results”.
Again, we gladly adopted this change.
We would like to truly thank you for your time, your great efforts and the critical review of our manuscript! It has certainly helped to improve our work.
Reviewer 3 Report
This is an interesting retrospective study by Rugg et al regarding the potential beneficial use of hemoabsorption via Cytosorb in septic shock patients. The authors report that Cytosorb had a significant effect on patient survival even though the baseline characteristics of the patients that received Cytosorb therapy were worse than the control group regarding initial SOFA score and norepinephrine requirements. The authors also identify risk factors for mortality within the Cytosorb group (namely lactate levels and low platelet count). The study tries to provide further evidence for the use of specific hemoabsorption devices in septic shock patients.
Major issues
- Although the authors have been extremely careful in reporting the limitations of their study, and the conclusions derived are potentially sound, there is a main concern regarding the unusual characteristics of the control group and how data comparison was organized.
- It is highly unexpected that from 2936 ICU patients only 84 (2.86%) had a diagnosis of septic shock that needed RRT. Furthermore, it is also quite unexpected that from all these patients, there were 42 Cytosorb treated patients and the only eligible 42 non- Cytosorb treated patients were matched with a 1:1 ratio to the treated ones and with compatible baseline characteristics. I would expect the septic shock patients requiring RRT to be more. In such a case a 1:3 (treatment : control) design with proper baseline characteristics matching would be more reasonable.
- There are also several other indications that the selection of controls could is an issue in this study, including the data reported in Figure 1 (page 6).
- In the latter figure, there is also an indication that the two groups are not comparable since 0 is defined as “time of Cytosorb mounting in the Cytosorb group and time to ICU admission in the Match group”.
- Page 4, lines 33-36. Please clarify the range of Cytosorb mounting after ICU admission since the range 0.5-719 means that in at least one patient that was included in the study the device was mounted on day 30 after admission. In combination with comment 1.c, this could mean that Cytosorb was also used in patients with secondary/ICU-related septic shock (after their initial septic shock diagnosis that led to their admission). These two groups of patients are potentially non-comparable in a clinically meaningful manner.
- It would also be crucial to clarify issues of potential outliers and differences in the concurrent presence of multiple comorbidities between the two groups.
Therefore, there is a clear risk for selection bias that cannot be fully evaluated by a reviewer with the provided data but could be condemnatory for the study conclusions.
- Page 3 lines 18-21, Assumption of a GCS 5/15 for all patients due to lack of complete documentation is highly irregular and (in combination with the comment above) could affect results.
Minor issues
- Some English editing is needed throughout the manuscript
- There are two sets of different figures numbered 1 to 3 and within text reference to them needs to be corrected (e.g. in the bottom of page 5 figure 3A probably refers to figure 2 and 3 at the end of the manuscript in two appearances)
Author Response
Major issues
- Although the authors have been extremely careful in reporting the limitations of their study, and the conclusions derived are potentially sound, there is a main concern regarding the unusual characteristics of the control group and how data comparison was organized.
Dear Reviewer,
To begin, we would like to truly thank you for your time and your critical review.
We are well aware of methodical concerns with our study. Hence the extensive limitations section and carefully phrased conclusions.
For a better understanding in general, We would like to step by step explain to you the process of our data acquisition and analysis.
First, we retrospectively searched for patients treated with CytoSorb for septic shock and found the presented 42 patients. Initiation of CytoSorb therapy varied from 0.5 to 719 hours after ICU admission, but the majority of patients received treatment within the first days (n= 20 on day 1, n=7 on day 2, n=6 on day 3) leading to a median time of 21.4 hours after ICU admission. Once again, all these patients received CytoSorb due to upcoming septic shock regardless of the time of initiation. Given these facts, the question arose how adequate matching should take place. Particularly, since we sought to not only compare baseline characteristics and outcome but also the course of certain variables during ICU stay (including catecholamine requirements and organ failure; Figure 1 and A2 and A3).
We then decided to define baseline for the CytoSorb patients as the day of CytoSorb mounting. In order to unitise the matched controls, we decided to define their baseline to be the time of ICU admission and searched for matched controls as deteriorated on ICU admission as the CytoSorb patients were just before CytoSorb initiation. Due to missing data on admission diagnoses and catecholamine requirements within our primary database, matching initially took place for age, gender, lactate, lab components of the SOFA score (creatinine, thrombocytes, bilirubin) and lab values of infection (leukocytes, crp, pct). In a second step, potential matches were assessed for an admission diagnosis of septic shock and requirement of RRT.
Simply speaking, we compared patients receiving CytoSorb therapy for septic shock (at any time during ICU stay, but mainly within the first days) with septic shock patients requiring RRT and presenting as deteriorated on ICU admission as the CytoSorb patients did just before CytoSorb initiation.
- It is highly unexpected that from 2936 ICU patients only 84 (2.86%) had a diagnosis of septic shock that needed RRT. Furthermore, it is also quite unexpected that from all these patients, there were 42 Cytosorb treated patients and the only eligible 42 non- Cytosorb treated patients were matched with a 1:1 ratio to the treated ones and with compatible baseline characteristics. I would expect the septic shock patients requiring RRT to be more. In such a case a 1:3 (treatment : control) design with proper baseline characteristics matching would be more reasonable.
We fully agree to your statement!
But as mentioned above, due to the chosen study design, matching primarily took place without knowledge of the underlying admission diagnosis. The total number of treated septic shock patients in our department is not referred to in our manuscript (Figure A1 in the appendix).
Furthermore, the day of CytoSorb initiation (lab values closest to CytoSorb intiation) was matched to the day of ICU admission (admission lab). The number of patients who were as deteriorated directly on ICU admission as CytoSorb patients were directly before CytoSorb initiation were limited. Especially because those deteriorated due to septic shock in this manner commonly would receive CytoSorb in our department. To overcome this numerical limitation, we extended our search for patients back to 2011 – a timeframe where CytoSorb was not available yet.
Septic shock patients treated in our department were certainly much more than 84 in the analysed timeframe. But septic shock patients so deteriorated directly on ICU admission but then again not receiving CytoSorb if they were, were hard to find.
In the end our high demands to a sufficiently comparable matched group, without losing patients in the treatment group, hindered the extension to a 1:n matching process.
- There are also several other indications that the selection of controls could is an issue in this study, including the data reported in Figure 1 (page 6).
- In the latter figure, there is also an indication that the two groups are not comparable since 0 is defined as “time of Cytosorb mounting in the Cytosorb group and time to ICU admission in the Match group”.
Dear Reviewer,
We do understand your objections to Figure 1.
First of all, catecholamine requirements differed significantly between the groups. Unfortunately, catecholamine requirements were not included in the matching process due to missing values in our primary database. Not until secondary analysis of the study population’s ICU PDMS these values were retrieved. As displayed in Table 1 and also mentioned throughout the manuscript the patients treated with CytoSorb had significantly increased catecholamine requirements (and SOFA Scores) at baseline when compared to the matched controls. Variables matched for (CRP, PCT, Leukocytes, Thrombocytes, Bilirubin, Creatinine, age, gender) and also other baseline values (sepsis source, comorbidities) were highly comparable, justifying our further analysis.
Secondly, while catecholamine requirements were mainly available for CytoSorb patients prior to CytoSorb initiation, they were not in the matched controls as their baseline was the time of ICU admission. Consequently, we could have resigned from depicting the timeframe before time 0 (-24h – 0h) also in the CytoSorb group. But instead, we decided to keep these values as they display some important information on the rise of catecholamine requirements prior to CytoSorb initiation opposed to their decrease thereafter.
- Page 4, lines 33-36. Please clarify the range of Cytosorb mounting after ICU admission since the range 0.5-719 means that in at least one patient that was included in the study the device was mounted on day 30 after admission. In combination with comment 1.c, this could mean that Cytosorb was also used in patients with secondary/ICU-related septic shock (after their initial septic shock diagnosis that led to their admission). These two groups of patients are potentially non-comparable in a clinically meaningful manner.
As mentioned before, it is true that some patients received CytoSorb therapy after already being on ICU for a while (n=1 on ICU day 4,5,7,9,12,27 and 31, n=2 on ICU day 8). Inclusion criteria was septic shock requiring RRT on ICU, regardless of sepsis source and time of outbreak.
Due to the matching process matched controls were unitised to sepsis on ICU admission.
We do understand your thoughts on non-comparability but want to emphasise that both groups included patients with primary and secondary sepsis - even when CytoSorb was mounted on ICU day 1! Admission diagnoses were not restricted to any kind of sepsis. Amongst them are patients with hospital and community acquired pulmo-sepsis or uro-sepsis, or primary or secondary (postoperative) abdominal sepsis and so on. As shown in Table 1 sepsis source and suspected pathogens were comparable between the groups with nearly half being gram negative infections. Greater intergroup differences with regard to hospital or community acquired infections can be out ruled as different pathogens would be expected.
In general, it is the diagnosis of sepsis per se which is incredibly heterogeneous.
- It would also be crucial to clarify issues of potential outliers and differences in the concurrent presence of multiple comorbidities between the two groups.
Dear Reviewer,
Thank you for outlining this very important point. In order to clarify this issue, we decided to add a row to Table 1 including a group comparison of “Total number of comorbidities” for all patients, survivors and non-survivors. Unsurprisingly, non survivors suffered from more comorbidities than survivors did but there were no differences between the CytoSorb and the Matched group with regard to all patients, survivors and non-survivors.
The following row was added to Table 1 (page 5):
CytoSorb group. Matched group
Median (IQR) Median (IQR) p
|
Total number of comorbidities |
|
|
|
The following sentence was added to the results section page 4 line 26:
“Particularly with regard to the total number of pre-existing comorbidities per patient, non-survivors suffered from more comorbidities than survivors did in both groups but there were no detectable differences between the groups.”
Therefore, there is a clear risk for selection bias that cannot be fully evaluated by a reviewer with the provided data but could be condemnatory for the study conclusions.
- Page 3 lines 18-21, Assumption of a GCS 5/15 for all patients due to lack of complete documentation is highly irregular and (in combination with the comment above) could affect results.
Dear Reviewer,
It is true that a GCS of 15/15 was assumed for all patients. This approach is in line with the cited reference from Lambden et al. in 2019 published in Crit Care:
Lambden, S.; Laterre, P.F.; Levy, M.M.; Francois, B. The SOFA score—development, utility and challenges of accurate assessment in clinical trials. Crit. Care 2019, 23, 374, doi:10.1186/s13054-019-2663-7.
They conclude the following with regard to GCS:
“The CNS component of the SOFA score is the least accurately measured and associated with the most errors.”
They state the following proposal: “The GCS value will be carried over from the last pre-intubation GCS throughout the duration of hypnotic/sedative medication administration. if: GCS from before intubation is not available, a value of 15/15 will be recorded and carried over throughout the duration of hypnotic/sedative medication administration.”
Or:
“Formal assessment of GCS can be undertaken from 24 h after the cessation of sedative medication by infusion. if: The clinician at the bedside is satisfied that the assessment is not affected by ongoing effects of sedative/hypnotic therapy.”
To be honest, we feel that simply carrying over the last pre-intubation GCS throughout sedation to be at least as inaccurate as assuming a value of 15/15, and probably as inaccurate as assessing a GCS after 24h of sedation cessation.
Minor issues
- Some English editing is needed throughout the manuscript
Dear Reviewer,
We proofread the manuscript and corrected to the best of our knowledge!
Thank you very much for your support in improving our manuscript.
- There are two sets of different figures numbered 1 to 3 and within text reference to them needs to be corrected (e.g. in the bottom of page 5 figure 3A probably refers to figure 2 and 3 at the end of the manuscript in two appearances)
Dear Reviewer,
There are two sets with each three Figures. Three in the main text (Figures 1-3) and three in the Appendix. Apparently, the figure-titles in the Appendix lost their numeration. We renamed them accordingly to the references in the main text to Figures A1, A2 and A3.
Thank you very much for your critical reading. We hope to have clarified many issues!
With best regards.
Round 2
Reviewer 3 Report
I would like to thank the authors for the detailed response to my previous remarks. They have provided a significant amount of information that is critical for a better understanding of their study. In such a "difficult" study, only limited and very low strength "preliminary" conclusions may be made.
I, therefore, believe that the details provided in their response to my initial review should be clearly included in the manuscript. This can be done by incorporating this information in the Methods section and in the limitations section, while some of it could be presented as supplemental data. In this way, readers will have the capacity to fully understand the limitations of the study and critically evaluate its results.
My minor comments have been fully addressed.
Author Response
REVIEWER:
I would like to thank the authors for the detailed response to my previous remarks. They have provided a significant amount of information that is critical for a better understanding of their study. In such a "difficult" study, only limited and very low strength "preliminary" conclusions may be made.
I, therefore, believe that the details provided in their response to my initial review should be clearly included in the manuscript. This can be done by incorporating this information in the Methods section and in the limitations section, while some of it could be presented as supplemental data. In this way, readers will have the capacity to fully understand the limitations of the study and critically evaluate its results.
My minor comments have been fully addressed.
Dear Reviewer,
Once more we must thank you for your time spent with our manuscript!
We do agree that the results of our study require a proper and above all transparent presentation of the applied methods and an in-depth mentioning of all limitations.
We went through our previous review report and implemented our answers to you within the manuscript.
We truly believe that the quality of our manuscript has substantially improved in doing so and are tremendously grateful for your support!
The following has been added:
In the Methods section:
The complete subsection “2.2. Study population” was adopted:
Page 3 line 1-32:
“Patients treated with CytoSorb between January 1, 2015 and December 31, 2019 were obtained from our documentation database. CytoSorb treatment is mainly but not exclusively indicated for septic shock in our department and is always combined with renal replacement therapy. By including patients only receiving CytoSorb treatment due to septic shock n= 42 patients were enrolled. Inclusion criteria was septic shock requiring RRT on ICU, regardless of sepsis source and time of outbreak. Admission diagnoses were not restricted to any kind of sepsis, leading to included patients suffering from primary as well as secondary sepsis. Initiation of CytoSorb therapy varied from 0.5 to 719 hours after ICU admission, but the majority of patients received treatment within the first days (n= 20 on day 1, n= 7 on day 2, n= 6 on day 3, n= 1 on day 4, 5, 7, 9, 12, 27 and 31, n= 2 on day 8) leading to a median time of CytoSorb mounting of 21.4 hours after ICU admission. As we sought to not only compare baseline characteristics and outcome but also the course of certain variables during ICU stay (including catecholamine requirements and organ failure), baseline for the CytoSorb patients was defined as the day of CytoSorb mounting. In order to unitise the matched controls, we decided to define their baseline to be the time of ICU admission and searched for matched controls as deteriorated on ICU admission as the CytoSorb patients were just before CytoSorb initiation. Matched controls were treated at our department for septic shock and required RRT but did not receive CytoSorb therapy. Again, admission diagnoses were not restricted to primary or secondary sepsis of any source. Anticipating a low number of patients as deteriorated directly on ICU admission as CytoSorb patients were directly before CytoSorb initiation the searched treatment timeframe for the control group was extended back to 2011, where CytoSorb was not available yet. Despite this extension the number of matching controls were limited, rendering an enlarged 1:n matching process impossible.
Data were obtained from a database containing all patients treated at our department within a timeframe from November 1, 2011 to December 31, 2019 (n= 2936). The primary data in this study contained basic demographics and laboratory values but did not include diagnoses nor catecholamine- or oxygen requirements or sequential organ failure assessment (SOFA) scores. Therefore, matching took place for age, gender, baseline-bilirubin, -creatinine, -c-reactive-protein, -procalcitonin, -lactate, -thrombocyte- and -leukocyte-count. In a second step the resulting matches were analysed for an admission diagnosis of septic shock and the requirement of renal replacement therapy in the local hospital information system. Matches missing above mentioned criteria were discharged and the matching process repeated for the remaining patients. The control group allocation process is depicted in Figure A1.”
The last passage was moved from the subsection “2.4. Statistics” as it was felt to be more appropriate here.
Within the subsection 2.3. Data acquisition:
Page 3 line 46- 48: “Due to incomplete documentation of Glasgow-Coma-Scales (GCS) prior to sedation and intubation as well as during ICU stay, a value of 15/15 was suspected throughout the study population, as also proposed by Lambden et al. [41].”
With regard to your comments on Figure 1 the following was added:
in the Results section “3.2. Development of catecholamine requirements and organ failure”
page 6 line 5-6: “Due to the study design, catecholamine requirements before baseline (ICU admission vs. CytoSorb initiation) were not available for the control group.”
The Limitations section already included the following:
Page 12 line 19-21: “Lastly, although both groups were comparable with regard to demographics, most baseline characteristics and disease aetiology, differences were still present with higher initial SOFA-scores and catecholamine requirements in the CytoSorb group.”
The Limitations section was extended by the following:
Page 12 line 4 -12: “Thirdly, our analysis included patients suffering from community and hospital acquired as well as primary and secondary sepsis. Particularly within the CytoSorb group patients were included with treatment initiation far beyond ICU admission. Comparability is therefore potentially hindered. In this regard, we must state that, inclusion criteria were not restricted to any kind of sepsis in neither group. ICU admission alone does not necessarily indicate any particular sepsis type. Most importantly, sepsis source and suspected pathogens were comparable between the groups with nearly half being gram negative infections in both groups. Greater intergroup differences with regard to hospital or community acquired infections can again be out ruled as different pathogens would be expected. Fourthly,…”
The Limitations section already included a passage on the different baselines, the matching process and the impossibility of a 1:n matching, and the different treatment timeframes (2011 vs. 2015).
Inclusion of your comments on multiple comorbidities were included in the first review round.